# Environmental and Psychosocial Barriers Affect the Active Commuting to University in Chilean Students

**DOI:** 10.3390/ijerph18041818

**Published:** 2021-02-13

**Authors:** Antonio Castillo-Paredes, Natalia Inostroza Jiménez, Maribel Parra-Saldías, Ximena Palma-Leal, José Luis Felipe, Itziar Págola Aldazabal, Ximena Díaz-Martínez, Fernando Rodríguez-Rodríguez

**Affiliations:** 1Grupo AFySE, Investigación en Actividad Física y Salud Escolar, Escuela de Pedagogía en Educación Física, Facultad de Educación, Universidad de Las Américas, Santiago 8370035, Chile; 2Área Salud, Universidad Tecnológica de Chile INACAP, La Serena 1700000, Chile; ninostrozajimenez@gmail.com; 3Magíster en Nutrición para la Actividad Física y el Deporte, Escuela de Nutrición y Dietética, Facultad de Ciencias, Universidad Mayor, Santiago 8580745, Chile; 4IRyS Research Group, School of Physical Education, Pontificia Universidad Católica de Valparaíso, Valparaíso 2374631, Chile; maribel.parra@pucv.cl (M.P.-S.); ximena.palmaleal@gmail.com (X.P.-L.); fernando.rodriguez@pucv.cl (F.R.-R.); 5School of Sport Sciences, Universidad Europea de Madrid, 28670 Madrid, Spain; joseluis.felipe@universidadeuropea.es (J.L.F.); itziar.pagola@universidadeuropea.es (I.P.A.); 6Quality of Life Research Group in Different Populations, Department of Education Sciences, Universidad del Bíobío, Chillan 3800949, Chile; xdiaz@ubiobio.cl

**Keywords:** active, commuting, active transport, physical activity, active behavior, college

## Abstract

Biking and walking are active commuting, which is considered an opportunity to create healthy habits. Objective: The purpose of this study was to determine the main environmental and psychosocial barriers perceived by students, leading to less Active Commuting (AC) to university and to not reaching the Physical Activity (PA) recommendations. Material and Methods: In this cross-sectional study, 1349 university students (637 men and 712 women) were selected. A self-reported questionnaire was applied to assess the mode of commuting, PA level and barriers to the use of the AC. Results: Women presented higher barriers associated with passive commuting than men. The main barriers for women were “involves too much planning” (OR: 5.25; 95% CI: 3.14–8.78), “It takes too much time” (OR: 4.62; 95% CI: 3.05–6.99) and “It takes too much physical effort “ (OR: 3.18; 95% CI: 2.05–4.94). In men, the main barriers were “It takes too much time” (OR: 4.22; 95% CI: 2.97–5.99), “involves too much planning” (OR: 2.49; 95% CI: 1.67–3.70) and “too much traffic along the route” (OR: 2.07; 95% CI: 1.47–2.93). Psychosocial barriers were found in both sexes. Conclusions: Psychosocial and personal barriers were more positively associated with passive commuting than environmental barriers. Interventions at the university are necessary to improve the perception of AC and encourage personal organization to travel more actively.

## 1. Introduction

Sedentary lifestyle represents an important risk factor for health, since it participates in the development of chronic non-communicable diseases such as cardiovascular diseases, type 2 diabetes and some types of cancer [1,2]. Sedentarism constitutes one of the main causes of mortality worldwide, especially among those who fail to comply with the recommendations of physical activity (PA) for the adult population [3].

This decrease in PA has not only been related to weight gain and worse psychological well-being [4], but also contributes to an increased risk of developing non-communicable diseases and lower life expectancy [5].

Active commuting (walking or cycling from one place to another), is considered an opportunity to create healthy habits, which improve PA levels, decrease cardiovascular risk [6,7] and help achieve a healthier body composition in the young and adult population [8]. Although there is limited evidence in developing countries about the association between active commuting (AC) and health benefits [9,10], Steell et al. [11] showed that, in Chile, 30 min of AC was associated with less adiposity and a healthier metabolic profile that includes a lower risk of obesity, diabetes and metabolic syndrome. Another prospective study, which included a cohort of 263,540 participants from the United Kingdom (Biobank), reported that AC on a bicycle was associated with a lower risk of cardiovascular disease, cancer and all-cause mortality [12].

The university stage is a period of transition from adolescence to adulthood, which is characterized by long days of study, a high sitting time [13], little time for PA [14] and generally bad eating habits [15]. The decrease in PA [16] is mainly due to the fact that the subject of physical education is not mandatory in university, as it is at school [17]. Likewise, sports practice improves physical self-concept, which improves physical appearance, physical ability and weight control behaviors [18]. In Chile, it has been shown that men and women who perform sports activities have a more positive self-concept as compared to men and women who do not perform sports activities [14]. This positive physical self-concept could be associated with lower barriers to PA and AC, but this has not been studied to date.

Studies have shown that the main barriers to PA among university students are related to a lack of time, lack of social support, lack of motivation/enjoyment and economic reasons [19,20,21]. PA behaviors are influenced by personal (knowledge, skills, attitudes, and self-esteem) and environmental factors (social support, institutional characteristics and built environment [22,23]. However, there is no clarity as t which barriers affect PA in university students, especially considering the different dynamics of different universities [24].

In the same way, the choice of AC, such as walking, can be influenced by several barriers, such as distance and socioeconomic status. It was observed in a study in Spanish university students that those who lived less than 2 km (km) away from the university and those who had a low socioeconomic status used to walk [25]. A short travel distance, high connectivity on the streets, living in an urban area and high density on the roads have been positively associated with higher levels of AC [26,27].

In order to follow the World Health Organization (WHO) recommendation that urban planning in cities promote PA and AC through the design of urban spaces [28], it is important to analyze the previous patterns of population commuting [29,30] to achieve the implementation of promotion programs, improvements in bikeway and walking infrastructure, and road safety [31].

According to the evidence presented and the importance of increasing the level of PA in university students by active commuting, it is necessary to get an idea of the mode of commuting in Chilean university students. The objective of this study was to determine the main environmental and psychosocial barriers perceived by students and associated with less AC to university and not reaching the PA recommendations.

## 2. Materials and Methods

### 2.1. Study Design and Participants

This cross-sectional study had a non-probabilistic sample (intentional) and had a descriptive and correlational analysis in university students from three Chilean regions.

A total of 1349 students (637 men and 712 women), with an average age of 22.7 ± 5.8 years, from three public (two in Valparaíso, Viña del Mar and one in Chillán) and one private university (Santiago) were selected. They all agreed to participate voluntarily in the study. They were regular students from the first to fifth academic year studying in fields such as Education (*n* = 265), Health (*n* = 308), Engineering (*n* = 716) and Social Sciences (*n* = 60). More information and inclusion criteria are shown in Figure 1.

### 2.2. Instruments

A self-reported questionnaire was applied to assess the mode of commuting [32], PA level [33] and barriers to the use of the AC in university students [34]. The instrument consisted of five items: sociodemographic characteristics (14 questions); mode of commuting (14 questions); barriers to AC (14 questions); PA level (4 questions) and a self-assessment questionnaire on physical condition—IFIS (International Fitness Scale) (not included in present analysis). This instrument was previously subjected to a specific reliability process for Chilean university students [35] In order to evaluate the reliability of this questionnaire, a test–retest process was performed. Kappa coefficient and Intraclass Correlation Coefficient (ICC) were calculated. Commuting to and from university was found to be in almost perfect agreement, with Kappa coefficient values of 0.882 and 0.822, respectively. ICC scores on distance to and from university and time to and from university showed good reliability in all its items, with high according values.

The sociodemographic characteristics item was sex, age, field of study (*health*, *education*, *social science*, *engineer*), university (*public*, *private*), residence area (*urban*, *rural*), live with family and socioeconomic level. To determine socioeconomic level, the Family Affluence Scale (FAS) was used, with the following questions: *(1) “Does your family own a car?”* (No = 0; Yes, one = 1; Yes, two or more = 2), *(2) “How many computers does your family own?”* (None = 0; One = 1; Two = 2; More than two = 3), *(3) “Do you have your own bedroom for yourself?”* (No = 0; Yes = 1) and *(4) “Do you have internet access?”* (No = 0; Yes = 1). Each answer was summed to obtain the total points. A score was assigned, and participants were classified into three categories regarding the FAS: low (0–3 points), medium (4–5 points) and high (6–7 points) [36].

The mode of commuting to university was defined with the questions: *(1) “How do you usually get to university?”* and *(2) “How do you usually get home from university?*”, with a choice of answer options such as walking, bicycle, car, motorcycle, public bus, metro/train and other modes. Walking and bicycle were categorized as “active” modes and other motorized modes as “passive” commuting. The items pertaining to mode of commuting and barriers to the use of AC were used.

The barriers to AC were indicated, such as “There are no sidewalk or bikeways”; “Bikeways occupied by people who walk”; “There is too much traffic along the route”; “Are dangerous crossing along the way”; “Walking or biking is insecure due to crime”; “Are no places to leave the bicycle safely”; “Streets are dangerous because of the cars”; “I get hot and sweat when I’m walking or biking”; “I’m too loaded to go walking or cycling”, “It is easier to move with car or motorcycle”; “Walk or biking involves too much planning”; “It takes too much time”; “It takes too much physical effort”; “I need the car or the motorcycle to university”. These questions had a categorical like-type response with the alternatives: totally agree = 1; agree = 2; disagree = 3; and strongly disagree = 4 [34].

The International Physical Activity Questionnaire (IPAQ, short version) was used to determine PA levels [33,37]. The PA was classified into sedentary time (min/day), light PA (min/week), moderate PA (min/week), vigorous PA (min/week) and moderate-vigorous (MVPA) as has been reported in a previous study [16]. Regarding the MVPA recommendations for adults (≥150 min/week) [38], students were classified as “Reaching” or “Not reaching” the weekly recommendations.

### 2.3. Procedure

With the authorization of the academic direction of each university and the knowledge of the career directors, the questionnaire was applied in paper format.

The application of the questionnaire was carried out from Monday to Friday in the day, evening and executive programs, in the same classroom. The application was carried out in 15–20 min and between the months of April and July 2017. Informed consent was obtained from each student before their participation, which requested the authorization to participate in the research project and explained the objectives and that the collected data were anonymous, private, confidential and for exclusive use in the study.

All the participants voluntarily agreed to participate in the study, which was approved by the Ethics Committee of the corresponding university (Code: CCF02052017) and governed by the Declaration of Helsinki 2013 [39].

### 2.4. Statistical Analysis

The results are presented in frequencies for categorical variables and for continuous variables, in means and standard deviations (M ± SD). To establish the associations between barriers and mode of commuting and barriers and compliance with MVPA, a binary logistic regression was applied to obtain the Odds Ratio (OR) and Confidence Interval (95% CI). Mode of commuting was included in the model as the dependent variable and barriers were included individually as independent variables. The score was also calculated for the barriers, which were grouped into the categories of Environment and Psychosocial. Significant values of *p* < 0.05 were considered. For the analyses, the statistical software IBM SPSS, version 26, was used.

## 3. Results

Table 1 present the description and sociodemographic characteristics of the participants. The distribution of the sample by sex was similar (52.8% women and 47.2% men). In both sexes, there was a higher percentage of students in the age range from 18 to 24 years (79.2%), belonging to the engineering study field (53.1%), who reside in urban areas (96.1%), live with their families (73.41%), and have a medium socioeconomic level (52.9%).

Table 2 shows the mode of commuting of university students according to sex. A total of 82.2% commuters were passive commuters and 17.8% were active commuters, with men (33.8%) forming a higher percentage of those who moved actively. In both sexes, the main mode of commuting was by public bus. The proportion of women and men who reported walking to university was 16.9% and 32%, respectively, while bicycle use was 1% and 1.7%, respectively.

Table 3 exhibits the PA of university students according to sex. According to the PA, the mean sitting min/day for the entire sample was 495.7 ± 696.9. Similar data were found in both sexes for minutes/day sitting (women 508.4 ± 751.7 and men 481.6 ± 630.2). Most of the sample (83.5%) did not comply with the recommendation of MVPA for 150 min a week.

Table 4 shows the association between barriers and passive commuting by sex. In women, the barriers *“Walk or biking involves too much planning”* (OR: 5.25; 95% CI: 3.14–8.78; *p* < 0.001), *“It takes too much time”* (OR: 4.62; 95% CI: 3.05–6.99; p < 0.001), *“It takes too much physical effort”* (OR: 3.18, 95% CI: 2.05–4.94, p < 0.001), were the main barriers. The perception of barriers *“The bikeways are occupied by people who walk”* (OR: 2.54; 95% CI: 1.69–3.84; *p* < 0.001), and *“I’m too loaded to go walking or cycling”* (OR: 2.44; 95% CI: 1.65–3.61; *p* < 0.001), is associated with the probability of passive commuting increasing by two times. In men, *“It takes too much time”* (OR: 4.22; 95% CI: 2.97–5.99; *p* < 0.001) increases the probability of choosing passive commuting four times, while *“Walk or biking involves too much planning”* (OR: 2.49; 95% CI: 1.67–3.70, *p* < 0.001) and *“There is too much traffic along the route”* (OR: 2.07; 95% CI: 1.47–2.93; *p* < 0.001) increases the probability of passive commuting two times. In general, women presented a greater association between barriers and passive commuting than men.

Table 5 indicates the association between barriers to AC and complying with the MVPA recommendation by sex. The statement “There are no sidewalk or bikeways” (OR: 1.81; 95% CI: 1.02–3.19; *p* < 0.05) was positively associated with non-compliance with the recommendations for MVPA in women. In addition, women who reported being “I get hot and sweat when I’m walking or biking” (OR: 0.56; 95% CI: 0.35–0.89; *p* < 0.05) as a barrier, and men who referred to “It takes too much physical effort” (OR: 0.66; 95% CI: 0.44–0.99; *p* < 0.05) as a barrier, are less likely to comply with the recommendations for MVPA, since the perception of these barriers is negatively associated with compliance with the recommendations.

## 4. Discussion

The main objective was to determine the barriers perceived by students to active commuting to university and the association with physical activity. The main findings were that the passive commuting was the most-used commuting mode to university in Chilean students. The most common barriers associated with passive commuting were “walk or biking involves too much planning” and “it takes too much time” in both sexes.

### 4.1. Mode of Commuting

The mode of commuting to university in the current study was mainly non-active commuting (82.2%). Similar results were reported in Spain, where a study of 518 students from two universities revealed that 65.1% of participants engaged in non-active commuting [19]. On the other hand, a study conducted at Kansas State University found a prevalence of 34.7% for non-active commuting and 65.3% for active commuting [40], a higher percentage compared to our study. In this regard, a study carried out by the Autonomous University of Barcelona justified the use of the non-active commuting mode for long distances between the university campuses, because the infrastructure was only available for motorized transportation [41]. In the United States, in a study conducted by the University of Kent, students were classified according to their place of residence (on the university campus or outside of it), revealing that only 4% of the students who lived off-campus walked, compared to 42% of students who lived on campus that walked, and 3% who cycled, highlighting the importance of distance in the choice of commuting mode [42]. In another study, also conducted in the United States, 76.1% of students actively moved [43]. These are high figures compared to this Chilean study, because students reside in different districts of the cities, since universities do not have on-campus housing. Thus, in our study, the main passive mode of commuting was public bus, with 50.5%, which was higher in women than in men. A Spanish study done on university students defined the use of the metro/train (31.1%) as the main non-active commuting mode [19], revealing the difference that exists in commuting modes compared with Chilean students. Public transport is more often classified as passive commuting [44]. However, there could be a small benefit associated with its use, since students usually have to walk to public bus stops [45]. In this sense, the choice of the mode of commuting to university is extremely important, not only for the benefits of AC, but also for the increase in daily PA.

### 4.2. Barriers Perception for Active Commuting

The perception of barriers was divided into “Environmental” and “Psychosocial”, and both variables had a significant association with the choice of commuting to university. In both sexes, the most often perceived barriers were “Walk or biking involves too much planning” and “It takes too much time”. In an Australian study of AC to and from university, travel time was the most important barrier to AC [46], which is consistent with our study. Another study conducted in university students in Ireland showed that an increase in travel time to university decreased the probability of being classified in a group containing AC and recreational PA at university [47]. This could indicate that students seek to minimize their commuting time, and that it is necessary to provide advice on travel planning and promote walking and cycling, especially for those who live near the university.

In our study, it was possible to appreciate that there were more barriers caused by the personal (psychosocial) compared to environmental barriers. A study carried out on university students from Spain showed that both the psychosocial and environmental variables had a significant correlation with AC to the university [48]. Another study in Spain on AC showed that socioeconomic factors are the most decisive with respect to the use of passive commuting, followed by social behavior variables [41]. A study in Africa on the effect of various motivators and barriers in cycling showed that addressing physical or environmental barriers individually has little impact on promoting cycling, as the perceived motivating variables were more personal [49]. In this sense, and according to the results of our study, it is important to take a comprehensive approach to psychosocial and environmental barriers, since the understanding of psychosocial barriers provides a useful framework to understand the mindset of travelers when designing policies that promote AC.

On the other hand, women showed a greater negative association between AC and the perception of environmental barriers compared to men. In a study on the influence of AC in college students, it was observed that men were more likely to use AC than women, and that, among women, there was a relationship between appearance (e.g., being sweaty) and AC [50]. In addition, this is in agreement with other studies that indicate that women are more concerned about access to services such as showers [51], and that clothing can play an important role in travel decision [52], which coincides with some perceived psychosocial barriers in this study. Another study on the sex gap in choosing to use bicycles showed that women choose bicycles 30% less often than men for their trips to campus and that there are various factors for not commuting by bicycle, such as an unsafe environment [53], which is also consistent with several environmental barriers perceived in this study by women. These findings reinforce the idea that the design of future interventions to promote AC should consider the specific barriers of women and men. Looking at the environmental and social factors that affect the perspective of women and men could directly contribute to increasing rates of AC.

In our study, it was possible to appreciate that there is a perception of both psychosocial and environmental barriers that would affect the use of bicycles as an active means of transport, with the former being the ones that have the greatest influence. Although it has been described that bicycle use not only depends on the individual mobility behaviors of the user but is also associated with the environment, urban cycling interacts with other modes of travelling, such as public buses, cars, motorcycles and pedestrians [54]. The use of a bicycle is associated more with the cyclist, as long as he/she controls the conditions of the trip, such as the distance travelled [55], physical effort and greater exposure to the weather, which conditions the mobility behaviors that stand out as individual, sociodemographic and psychological factors [56].

A study carried out in Argentina showed a lack of road safety when sharing the road with motorized vehicles as a main barrier to bicycle use [57]. In Chile, a study showed that the bicycle is used downtown through the streets, since bikeways are located in certain sectors of the metropolitan region (137 bikeways in 14 districts), which are mainly concentrated in districts of higher socioeconomic status and greater motorization [58]. As safety is an important factor in the choice of the bicycle as a means of AC, it would be important to promote changes in the cycling infrastructure that would make students perceive cycling as safer, such as improvements to local bike routes and the creation of more off-street bikeways. It is fundamental to improve the behaviors of PA practices through programs aimed at people that decrease the personal barriers to the use of this means of AC and to invest in road education.

In relation to the use of vehicles, an investigation showed that postgraduate students have a greater tendency to use passive modes of commuting (bus or car) due to their work or the possibility of acquiring a vehicle [59]. In Chile, the statistics show that the automotive fleet continues to increase [60], which could become a barrier with greater weight for AC. In our study, car use was only a barrier in men. However, in this university stage, students have low purchasing power; young people do not own cars and it is not presented as a significant barrier.

### 4.3. Barriers Perception for MVPA Recommendation

The barriers associated with non-compliance with MVPA are greater in women than in men. In women, in terms of environmental barriers, we find “There are no sidewalk or bikeways” and in the psychosocial barrier is “I get hot and sweat when I’m walking or biking”, compared to men who only have the psychosocial barrier “It takes too much physical effort”. It is important, at this point, to emphasize that the perceived barriers associated with MVPA coincide with AC, that is, a lot of effort and sweating are repeated reasons for women for not doing PA in this study.

In the literature, it was found that the barriers to compliance with the MVPA were economic levels, interest in the use of sports facilities, residence, intrapersonal and interpersonal barriers [22], distance [61] and psychosocial factors [47]. In the other study, carried out in university students, it is mentioned that the barriers to compliance with PA are economic levels, health, peer support, self-efficacy and effect related to the practice, which were increased in men compared to women [62]. In comparison with other investigations, women face greater barriers to the practice of PA [63].

On the other hand, Sevil et al. [64] analyzed the relationships between physical activity and the perceived barriers to physical activity, motivation and stages of change in Spanish university students, where they found that the barriers to participation were related negatively to the levels of PA and more self-determined forms of motivation. Recommendations include intervention from a medical area to help to comply with the recommendations of physical activity for optimal health [65], due to the high incidence of sedentary lifestyle in university students [66]. A study of Peruvian medical college students from a private university indicates that of the 312 students, just under a third performed MVPA for ≥150 min/week, and slightly more than a third performed MVPA for ≤30 min/week [67]. In turn, a study carried out with 244 adults mentions environmental barriers, where they indicate that cold days with little light are a barrier to compliance with the MVPA, because the increase of 10° is associated with an additional increase of 1.5 min/day and every extra hour of light in the day adds 2.23 min to the practice of PA [68]. Moreover, in an investigation in 507 adolescents, they mention internal and external barriers, such as “I am not interested in physical activity” and “I need equipment I don’t have”, and no significant differences between sex were found [69]. In this sense, a better understanding of the barriers that prevent compliance with the MPVA between both sexes is essential to minimize passive commuting and obtain the variety of benefits associated with AC.

### 4.4. Limitations and Implications

Despite the numerous significant findings in our study, there were some limitations, including that the student questionnaire was self-reported, and therefore could be subject to bias. On the other hand, the data used were only related to travel between home and university, and cannot be extrapolated to other environments. Our study only recorded the main mode used for commuting, so we did not consider mixed modes. Finally, only four universities were consulted in the research and the geographical characteristics of the three cities where they are located are heterogeneous, and a representative sample per university was not calculated. Therefore, precaution must be taken when generalizing the current results. However, this study makes a significant contribution to the literature on the variety of influences that can affect active commuting. A change in the culture of mobility must begin from within communities and in this sense, university students can be an important target group, especially women, who show a greater negative association between the perception of environmental barriers and AC, as well as with non-compliance with MVPA. Therefore, these findings, in combination with existing research, provide a solid foundation for future studies in this area and the development of policies and programs to improve AC in the university, suggesting that a strong association between the government and university organizations can produce positive results.

## 5. Conclusions

This study provided important information on perceived barriers to AC and MVPA compliance in Chilean university students. The results of the present study suggest that psychosocial or personal barriers were more positively associated with passive commuting to university than environmental barriers. In other words, aspects based on personal decision to commute actively intervene to a greater extent than the barriers imposed by the environment. Therefore, implementing policies that address the psychosocial factors more, but environmental factors as well, are necessary to increase AC rates, and thus achieve an impact in terms of both health and PA in university students. In addition, our study suggests that it is necessary to target women over men for AC and PA interventions, since women present more barriers and less active commuting. In this way, the measures should not only be applied by the government, but should mainly include universities as the lead actor for the development of educational strategies that promote and increase AC.

## Figures and Tables

**Figure 1 ijerph-18-01818-f001:**
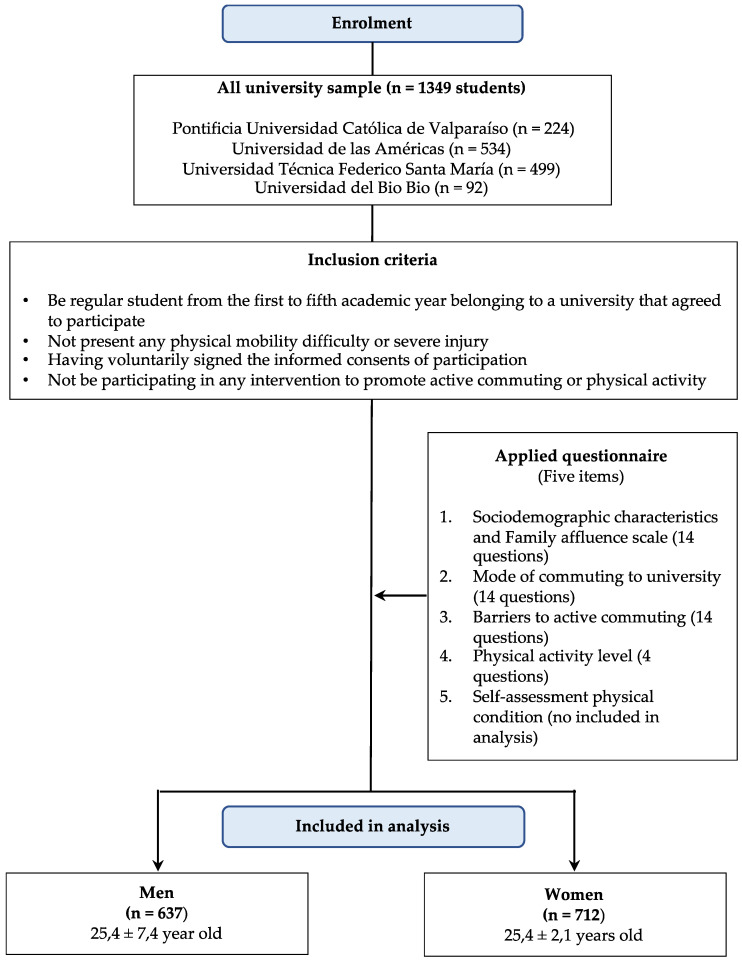
Methodological diagram to explain the sample and instrument.

**Table 1 ijerph-18-01818-t001:** Sociodemographic characteristics of the participants by sex.

Sociodemographic	All		Women		Men	
	*n*	(%)	*n*	(%)	*n*	(%)
Sex	1349	(100)	712	(52.8)	637	(47.2)
Age (years old)						
18–24	1041	(79.2)	669	(95.0)	372	(61.0)
25–31	159	(12.1)	32	(4.5)	127	(20.8)
32–38	69	(5.3)	3	(0.4)	66	(10.8)
39–46	32	(2.4)	0	(0)	32	(5.2)
47–54	13	(1.0)	0	(0)	13	(2.1)
Study field						
Health	308	(22.8)	216	(30.3)	92	(14.4)
Education	265	(19.6)	206	(28.9)	59	(9.3)
Social sciences	60	(4.4)	37	(5.2)	23	(3.6)
Engineer	716	(53.1)	253	(35.5)	463	(72.7)
Type of university						
Public	815	(60.4)	349	(48.9)	466	(73.1)
Private	534	(39.6)	363	(51.0)	171	(26.8)
Residence area						
Urban	1296	(96.1)	681	(95.6)	615	(96.5)
Rural	53	(3.9)	31	(4.4)	22	(3.5)
Live in family						
Yes	963	(71.4)	565	(79.4)	398	(62.5)
No	386	(28.6)	147	(20.6)	239	(37.5)
Socioeconomic level						
Low	12	(0.9)	9	(1.3)	3	(0.5)
Medium	713	(52.9)	373	(52.4)	340	(53.4)
High	624	(46.3)	330	(46.3)	294	(46.2)

**Table 2 ijerph-18-01818-t002:** Mode of commuting for the university students by sex.

	All		Women		Men	
	*n*	(%)	*n*	(%)	*n*	(%)
Mode of commuting						
Passive	1007	(74.6)	585	(82.2)	422	(66.2)
Active	342	(25.4)	127	(17.8)	215	(33.8)
Commuting to university						
Walking	324	(24.0)	120	(16.9)	204	(32.0)
Bike	18	(1.3)	7	(1.0)	11	(1.7)
Car	179	(13.3)	103	(14.5)	76	(11.9)
Moto	7	(0.5)	1	(0.1)	6	(0.9)
Public bus	681	(50.5)	407	(57.2)	274	(43.0)
Train/metro	140	(10.4)	74	(10.4)	66	(10.4)

**Table 3 ijerph-18-01818-t003:** Physical activity and recommendations of the university students by sex.

	All		Women		Men	
	Mean	±SD	Mean	±SD	Mean	±SD
Physical activity						
Sedentary (min/day)	495.7	±696.9	508.4	±751.7	481.6	±630.2
Light PA (min/week)	262.3	±348.8	224.9	±316.6	304.2	±377.5
Moderate PA (min/week)	113.0	±186.3	100.5	±174.5	126.9	±197.8
Vigorous PA (min/week)	115.6	±160.1	73.6	±118.0	162.5	±185.9
Moderate-vigorous (MVPA)	78.9	±70.6	65.3	±68.0	94.2	±70.3
Recommendation weekly MVPA						
No reach [*n* (%)]	1127	(83.5)	627	(88.1)	500	(78.5)
Reach [*n* (%)]	222	(16.5)	85	(11.9)	137	(21.5)

Abbreviations: (mean ± SD) Mean ± Standard Deviation; (PA) Physical Activity (MVPA) Moderate-Vigorous Physical Activity.

**Table 4 ijerph-18-01818-t004:** Association between barriers and passive commuting by sex.

	Passive Commuting to University					
Barriers	Women			Men		
	OR	(CI 95%)	*p* Value	OR	(CI 95%)	*p* Value
Environmental						
There are no sidewalk or bikeways	1.27	(0.84–1.92)	0.264	0.87	(0.60–1.26)	0.446
Bikeways occupied by people who walk	**2.54**	**(1.69–3.84)**	**<0.001**	1.28	(0.91–1.78)	0.152
There is too much traffic along the route	**1.51**	**(1.00–2.27)**	**0.049**	**2.07**	**(1.47–2.93)**	**<0.001**
Are dangerous crossings along the way	**2.12**	**(1.36–3.31)**	**0.001**	**1.55**	**(1.09–2.22)**	**0.016**
Walking or biking is insecure due to crime	1.23	(0.83–1.84)	0.308	0.75	(0.53–1.04)	0.086
Are no places to leave the bicycle safely	1.04	(0.71–1.52)	0.850	1.02	(0.73–1.43)	0.898
Streets are dangerous because of the cars	**2.18**	**(1.39–3.40)**	**0.001**	**1.70**	**(1.17–2.45)**	**0.005**
Psychosocial						
I get hot and sweat when I’m walking or biking	**1.52**	**(1.03–2.24)**	**0.034**	**1.60**	**(1.14–2.23)**	**0.007**
I’m too loaded to go walking or cycling	**2.44**	**(1.65–3.61)**	**<0.001**	1.10	(0.79–1.53)	0.590
It is easier to move with car or motorcycle	1.38	(0.94–2.02)	0.104	**1.75**	**(1.26–2.44)**	**0.001**
Walk or biking involves too much planning	**5.25**	**(3.14–8.78)**	**<0.001**	**2.49**	**(1.67–3.70)**	**<0.001**
It takes too much time	**4.62**	**(3.05–6.99)**	**<0.001**	**4.22**	**(2.97–5.99)**	**<0.001**
It takes too much physical effort	**3.18**	**(2.05–4.94)**	**<0.001**	**1.86**	**(1.30–2.65)**	**0.001**
I need the car or the motorcycle to work	1.09	(0.72–1.66)	0.680	0.96	(0.68–1.37)	0.837

Significant association in bold as *p* < 0.05 and *p* < 0.001.

**Table 5 ijerph-18-01818-t005:** Association between barriers to active commuting and compliance with the MVPA recommendation by sex.

	Reach MVPA Recomendation					
Barriers	Women			Men		
	OR	(CI 95%)	*p* Value	OR	(CI 95%)	*p* Value
Environmental						
There are no sidewalk or bikeways	**1.81**	**(1.02–3.19)**	**0.042**	0.90	(0.59–1.36)	0.610
Bikeways occupied by people who walk	0.87	(0.55–1.37)	0.538	0.93	(0.63–1.36)	0.707
There is too much traffic along the route	1.03	(0.62–1.71)	0.918	1.30	(0.85–1.97)	0.225
Are dangerous crossings along the way	1.54	(0.79–3.00)	0.201	1.50	(0.96–2.35)	0.076
Walking or biking is insecure due to crime	0.95	(0.59–1.54)	0.839	0.83	(0.57–1.21)	0.332
Are no places to leave the bicycle safely	0.89	(0.57–1.40)	0.621	0.79	(0.54–1.17)	0.246
Streets are dangerous because of the cars	1.51	(0.78–2.94)	0.224	1.03	(0.67–1.60)	0.889
Psychosocial						
I get hot and sweat when I’m walking or biking	**0.56**	**(0.35–0.89)**	**0.014**	1.05	(0.72–1.54)	0.793
I’m too loaded to go walking or cycling	0.98	(0.60–1.58)	0.917	0.83	(0.57–1.22)	0.345
It is easier to move with car or motorcycle	0.79	(0.50–1.25)	0.311	1.29	(0.83–1.78)	0.312
Walk or biking involves too much planning	1.26	(0.80–1.98)	0.328	0.94	(0.62–1.42)	0.752
It takes too much time	1.15	(0.72–1.84)	0.566	0.76	(0.52–1.11)	0.159
It takes too much physical effort	0.94	(0.59–1.48)	0.780	**0.66**	**(0.44–0.99)**	**0.044**
I need the car or the motorcycle to work	0.74	(0.44–1.23)	0.239	0.85	(0.56–1.28)	0.424

Significant differences in bold were set at *p* < 0.05.

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
