# Peer review of "Environmental and Psychosocial Barriers Affect the Active Commuting to University in Chilean Students"

_ijerph, 2021, doi:10.3390/ijerph18041818_

Round 1

Reviewer 1 Report

This article is of relevant scientific and clinical value. Overall the manuscript, from Introduction to the Conclusion, is easy to read, the text is concise and logical and, conceptually and methodologically, correct. English language and style require minor spell check.

The authors did a good job in reviewing and using significant literature on the topic, including relevant and actual references.

Methods and results are fine, but I propose a revision of the title of table 2 and remove the PhysicalActivity variable from this table to a better understanding avoiding mixing (n/%) and (mean/SD).

Discussion and conclusions are written in fluent text. Good discussion, supported by pertinent references. Just two remarks: 1) Please refer the limitations of your study; 2) I would like to have read more about the implications for research & development on the topic.

Author Response

We appreciate the suggestions made to the article " Environmental and psychosocial barriers affect the active commuting to university in Chilean students”.

We have incorporated your comments that you have suggested, which have fed into the text. We have also made some changes to the comments that you have suggested. We will detail these changes in the table below, hoping to meet your expectations.

Reviewer 2 Report

The article is interesting to scientists and politics.
The introduction provides numerical information and there are abbreviations without explanation. This information is necessary to be clarified.
The objective of this study was to determine the barriers perceived by students at AC to university and its association with PA. However, the introduction reveals the importance of PA but does not comment on barriers of PA. The introduction does not substantiate the relevance and objectivity of this article.
The author did not provide a research tool (questionnaire). The links provided do not reveal the context of the data collected. Therefore it is difficult to evaluate the validity of this study.
I think the application of the research tool should be analyzed in more detail.
In the discussion, authors analyze mode of commuting; barriers perception for active commuting; barriers perception for MVPA recommendation. However, provided general conclusions, is not related to the issues under discussion.
I suggest adjusting the manuscript according to comments made. Sincerely.

Author Response

(The authors gave the same response as above.)

Reviewer 3 Report

The authors examined students' barriers to active commuting (AC) to university and its association with physical activity.
A self-reported questionnaire was applied to assess the mode of commuting, PA level and barriers for the use of the active commuting. This questionnaire is not an established questionnaire. Accordingly, the authors should have conducted a sample size analysis to estimate the number of participants before performing the study. The authors should also perform reliability analysis and Cronbach alpha estimation to examine the questionnaire's reliability and cohesion. Finally, the study refers to Chilean students, so it is impossible to generalize the results internationally.

Author Response

(The authors gave the same response as above.)

Reviewer 4 Report

The proposed manuscript is focused on the survey research in the area of Physical Activity (PA) and active commuting (AC) of students from Chile. The investigated issues adjust to the scientific scope of the Journal. The editorial requirements are satisfied. Moreover, all references are cited in the text. The proposed article needs minor revision before it can be accepted for publication. The main reviewer's suggestions are given below:

1. The abstract needs revision and adjusting to the Journal requirements. For example, the text should not include terms "objective", "methods", etc. but should describe the issues connected with these terms (see examplary published articles in this Journal)

2. The Introduction section - the aims and scope of the study should be clearly defined. Moreover, the contribution of this study should be defined.

3. Materials and Methods section, the research methodology should be presented in a graphical form summarizing all steps connected with instruments, procedure or statistical analysis performed.

4. point 2.2. - Instruments - this section needs better structure implementation. Maybe adding tables or figures for defining e.g. barriers would improve the quality of the paper. In its current form, this section is difficult to read because it contains much of the information contained in the short communication.

5. Did the authors analyse the minimum sample size for assessing the quality of the surveys (for the assumed level of confidence and population size)? What is the power of the test?

6. The conclusion section - should be extended to provide, e.g., exemplary policies for increasing AC rates, the limitations of the performed research studies, and the authors possible future research works.

Author Response

(The authors gave the same response as above.)

Round 2

Reviewer 2 Report

I would like to thank the authors for their effort and time to produce a specific manuscript. The manuscript presents interesting research and appropriate for the journal.
The article is a finished paper concept.
Sincerely.